# SEM Analysis and Micro-CT Evaluation of Four Dental Implants after Three Different Mechanical Requests—In Vitro Study

**DOI:** 10.3390/ma17020434

**Published:** 2024-01-16

**Authors:** Ana Sofia Vinhas, Filomena Salazar, José Manuel Mendes, António Sérgio Silva, Blanca Ríos-Carrasco, José Vicente Ríos-Santos, Javier Gil, Mariano Herrero-Climent, Carlos Aroso

**Affiliations:** 1Oral Pathology and Rehabilitation Research Unit (UNIPRO), University Institute of Health Sciences (IUCS), CESPU, Rua Central de Gandra, 4585-116 Gandra, Portugal; ana.vinhas@iucs.cespu.pt (A.S.V.); filomena.salazar@cespu.pt (F.S.); jose.mendes@iucs.cespu.pt (J.M.M.); asergio.silva@iucs.cespu.pt (A.S.S.); carlos.ribeiro@iucs.cespu.pt (C.A.); 2Department of Periodontology, School of Dentistry, Universidad de Sevilla, C/Avicena S/N, 41009 Sevilla, Spain; jvrios@us.es; 3Bioengineering Institute of Technology, Faculty of Medicine and Health Sciences, Universitat Internacional de Catalunya, 08195 Sant Cugat-Barcelona, Spain; xavier.gil@uic.cat; 4Porto Dental Institute, 4150-518 Porto, Portugal; dr.herrero@herrerocliment.com

**Keywords:** implant–abutment connection, implant–abutment axial displacement, implant–abutment wear, fatigue-test, mechanical stresses, SEM analysis, Micro-CT evaluation

## Abstract

Statement of problem: Implant-supported rehabilitations are an increasingly frequent practice to replace lost teeth. Before clinical application, all implant components should demonstrate suitable durability in laboratory studies, through fatigue tests. Objective: The purpose of this in vitro study was to evaluate the integrity and wear of implant components using SEM, and to assess the axial displacement of the implant–abutment assembly by Micro-CT, in different implant connections, after three distinct mechanical requests. Materials and methods: Four KLOCKNER implants (external connection SK2 and KL; and internal connection VEGA and ESSENTIAL) were submitted to three different mechanical requests: single tightening, multiple tightening, and multiple tightening and cyclic loading (500 N × 100 cycles). A total of 16 samples were evaluated by SEM, by the X-ray Bragg–Brentano method to obtain residual stresses, and scratch tests were realized for each surface and Micro-CT (4 control samples; 4 single tightening; 4 multiple tightening; 4 multiple tightening and cyclic loading). All dental implants were fabricated with commercially pure titanium (grade 3 titanium). Surface topography and axial displacement of abutment into the implant, from each group, were evaluated by SEM and Micro-CT. Results: In the manufacturing state, implants and abutments revealed minor structural changes and minimal damage from the machining process. The application of the tightening torque and loading was decisive in the appearance and increase in contact marks on the faces of the hexagon of the abutment and the implant. Vega has the maximum compressive residual stress and, as a consequence, higher scratch force. The abutment–implant distances in SK2 and KL samples did not show statistically significant differences, for any of the mechanical demands analyzed. In contrast, statistically significant differences were observed in abutment–implant distance in the internal connection implants Vega and Essential. Conclusions: The application of mechanical compression loads caused deformation and contact marks in all models tested. Only internal connection implants revealed an axial displacement of the abutment into the implant, but at a general level, a clear intrusion of the abutment into the implant could only be confirmed in the Essential model, which obtained its maximal axial displacement with cyclic loading.

## 1. Introduction

An implant-supported dental prosthesis is composed of an implant, an abutment, and a fixing screw. The instability of prosthetic rehabilitation, caused by the maladjustment of the screws, alters the distribution of occlusal forces during function, which increases the micro-movements and microgap of the implant–abutment interface. This situation exacerbates infiltration at the implant–abutment interface causing biological complications, such as mucositis and peri-implantitis, which can ultimately lead to implant failure, with implications for professionals and patients [1,2].

In recent years, the geometries of implant connections have been developed with different biological and aesthetic characteristics. Two basic geometries are available: internal (INT) and external (EXT) connection. EXT connections typically have an external hexagon on the implant platform, while INT connections can be divided into internal hexagons, internal octagons, and cone Morse connections [3].

Mechanical complications, such as loosening and fracture of the prosthetic abutment fixation screw, have been associated with the type of implant–abutment connection [4]. This connection is the weakest point of the complex, as it must be resistant to occlusal forces, prevent micro-movement, and minimize bacterial micro-leakage [5].

In INT connections, the fixation and the stability of connection are not a function of the screw but rather conferred by the frictional resistance resulting from contact between the conical mating parts of the abutment and the implant. Frictional resistance is increased with the application of axial compressive forces [6]. The axial compressive forces may also increase the axial displacement of the abutment [7,8]. The axial displacement might be due to three factors: machining tolerance, settling effect, and wedge effect [9,10]. Machining tolerance, which is an intrinsic characteristic that exists between the machined implant components, contributes to the axial displacement of the components by dimensional variation and surface roughness [11]. The settling effect occurs between two different rough surfaces. When the abutment is placed in the implant, the settling effect occurs to varying degrees, by applying different torques to all systems. It increases depending on the micro-roughness between the metal surfaces of the abutment–implant connection [12]. The mechanism of the settling effect is based on the fact that there is no completely smooth surface, and settling occurs to soften the rough points, under pressure, because when the primary forces are used, these points are the only surfaces in contact. Machining tolerance and settling effect are inevitable phenomena in all implants [13].

Compared with EXT connections, INT connections have a wedge effect. In the wedge effect, the abutment acts as a wedge, concentrating the axial compressive force in the direction of abutment insertion, which increases the contact pressure and frictional resistance [6,14]. Moreover, as the tightening torque increases, the wedge effect of the abutment increases. It is considered that the axial displacement of the abutment, under functional loading, may have a great effect on the screw joint stability of an INT connection. When using the abutment, which lacks a vertical stop, the axial displacement of the abutment might occur under functional loading in the oral cavity. This implies that the tensional force within the screw is decreased, which indicates a loss of the preload. When the diminished preload reaches a critical point, under the loading, screw loosening will occur [15].

Axial displacement of abutments causes several mechanical problems in the implant–abutment complex, such as a decrease in the reverse torque of the screw after cyclic loading [16]. Negative occlusion may also occur, due to the abutment displacement, which leads to prosthesis displacement and therefore negative occlusion [9,10].

Screw initial preload is inserted by applying torque using a torque wrench. One of the main causes of screw loosening is “loss of preload”. Only 10% of the initial torque is transformed into preload, while the remaining 90% is used to overcome friction between surface irregularities [12,17,18].

The stability and integrity of the abutment–implant connection, by means of a screw, is fallible from the moment the prosthetic elements are joined and is dependent on the applied preload, wear of the components, and function. It is necessary to evaluate and quantify, with in vitro studies, the integrity of the structures of the system in the different connections.

To date, many researchers have investigated the surface deformation and wear of the connection areas, but there is a lack of studies that simultaneously focus on the wear of the components and the vertical displacement of the abutment in the implant, in the two main connection systems. 

The objectives of this study were as follows:

SEM evaluation of possible deformations and alterations on the surface of the implant–abutment union in the manufacturing state, after single tightening, multiple tightening, and multiple tightening and cyclic loading.

SEM evaluation of axial displacement of the abutment into the implant after cyclic loading, in internal and external connections.

Micro-CT evaluation of implant–abutment distance in the two connections after a single tightening, multiple tightening, and multiple tightening and cyclic load.

## 2. Materials and Methods

### 2.1. Description of Implants

A total of 4 Klockner (SOADCO-Andorra la Vella, Principat d’Andorra) implant connection systems were evaluated, including 2 internal connection systems (implant VEGA^®^ and implant ESSENTIAL^®^) and 2 external connection systems (implant SK2^®^ (with a hexagon of 1.8 mm) and implant KL^®^ (with a hexagon of 0.7 mm)). The implants used were 4 mm in diameter and 12 mm in length (Figure 1).

**Figure 1 materials-17-00434-f001:**
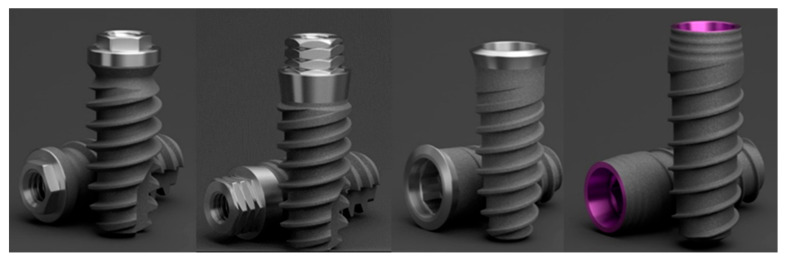
Klockner implants used: KL, SK2, Essential, and Vega.

### 2.2. Sample Size

The sample size was determined using the N Query Advisor program, based on the research of Cahsman (2011) [19]. With a significance of *p* < 0.05 and a power of 80%, the estimated sample size per group was N = 13 (rounded up to 15 in case of deterioration in any sample). This sample size was calculated in a previously published study to evaluate torque loss in different connections, after 3 different mechanical requests [20]. 

A total of 45 implants of each type were assessed, and placed in a metal support, leaving the most coronal portion 2 mm outside the device, respecting the ISO 14801:2016 standard [21] (Dentistry—Implants—Dynamic loading test for endosseous dental implants), which allowed the installation of the implants in the machine responsible for the application of the cyclic load as well as in the jaw of the device that evaluated the torque.

A total of 45 implants of each system were distributed in 3 groups of 15 each, to use in the different phases of the study, for a total sample of 180 implants.

A total of 45 straight titanium metal abutments for screw-retained prostheses from each platform were used in the study. These were supplied by the manufacturer from its usual commercial stock, along with the respective titanium fixing screws. In total, 180 abutments and 180 screws were used.

### 2.3. Mechanical Request

The 3 different types of mechanical requests were (Table 1):

U: single tightening. 

M: multiple tightening. 

MC: multiple tightening + cyclic loading. 

**Table 1 materials-17-00434-t001:** Sample identification table, with details of the codes used according to implant model and type of mechanical request.

Connection System	Mechanical Request	Sample Code
SK2^®^	Control	S
Vega^®^	Control	V
KL^®^	Control	K
Essential^®^	Control	E
SK2^®^	Single Tightening	SU
Vega^®^	Single Tightening	VU
KL^®^	Single Tightening	KU
Essential^®^	Single Tightening	EU
SK2^®^	Multiple Tightening	SM
Vega^®^	Multiple Tightening	VM
KL^®^	Multiple Tightening	KM
Essential^®^	Multiple Tightening	EM
SK2^®^	Multiple Tightening + Cyclic load.	SMC
Vega^®^	Multiple Tightening + Cyclic load.	VMC
KL^®^	Multiple Tightening + Cyclic load.	KMC
Essential^®^	Multiple Tightening + Cyclic load.	EMC

Phase I: The abutment was connected to the implant; the screw was fixed with the torque recommended by the manufacturer. This operation was performed only once. The removal torque value was evaluated after 1 min with a digital torque meter.

Phase II: The abutment was connected to the implant; the screw was fixed with the torque recommended by the manufacturer. This operation was performed 10 times with a time interval of 15 s between each tightening. The removal torque value was evaluated after 1 min with a digital torque meter.

Phase III: The abutment was connected to the implant; the screw was fixed with the torque recommended by the manufacturer. This operation was performed 10 times with a time interval of 15 s between each tightening. The removal torque value was evaluated, with a digital torque meter, after applying an axial load of 500 Newtons for 1000 cycles.

### 2.4. SEM Evaluation

The scanning electron microscope (SEM) scans a sample with a focused electron beam and obtains images with information about the topography and composition of the sample. It allows a superficial and exhaustive evaluation of components by acquiring high-resolution images, using the interactions produced between an incident electron beam and the surface to be analyzed. For this study, a field emission scanning electron microscope “Field Emission Scanning Microscope” FSEM model “JSM-7001F Scanning Microscope” was used, under potential conditions of 15 KV and an approximate working distance between 12 and 18 mm. This instrument is equipped with an EDS spectroscopy analysis probe “Energy-Dispersive X-ray Spectroscopy” of the OXFORD brand, model Xmax20, which allows the identification of chemical composition through the acquisition of the characteristic X-ray emission of each chemical element.

A total of 16 samples were evaluated by SEM, one sample from each of the 3 study groups, and a sample of each implant model in the manufacturing state. The selected sample was the one that obtained the highest tightening torque value, according to the previously obtained results [20]. 

The observations of the two components (abutment and implant) of the 4 groups of samples were carried out under the same conditions: power at 15 kV and distance of 17.5 mm. However, given the different geometry of the components, groups S and K were analyzed at 45 ± 1° inclination, while sample groups E and V were analyzed at 35 ± 1° inclination.

### 2.5. Residual Stress

Residual stresses were measured with a diffractometer incorporating a Bragg–Brentano configuration (D500, Siemens, Munich, Germany). The measurements were performed for the family of planes (213), which diffracts at 2θ = 139.5·°. The elastic constants of Ti in the direction of this family of planes are EC = (E/1 + n)(213) = 90.3 (1.4) GPa. Eleven Ψ angles, 0°, and five positive and five negative angles were evaluated. The position of the peaks was adjusted with a pseudo-Voigt function using appropriate software (WinplotR, free access online, 2012 version, Birbeck, UK)), and then converted to interplanar distances (dΨ) using Bragg’s equation. The d Ψ vs. sen2Ψ graphs and the calculation of the slope of the linear regression (A) were performed with appropriate software version 7.0 (Origin, Microcal, Northampton, MA, USA). The residual stress is: σ = EC(1/d0)A, where d0 is the interplanar distance for Ψ = 0° [22,23,24]. 

### 2.6. Scratch Test

Five scratches were performed with a diamond stylus with a spherical tip of 50 µm radius on five samples for each surface. The test machine was RTEC Instruments (STM5000, San Jose, CA, USA). The length of each scratch was 2 mm. The loading rate was between 0.5 and 1.5 N/mm and the final load was between 0.5 and 2 N. The horizontal displacement rate was fixed at 20 µm/s. The critical normal force (critical load) at which adhesive failure was first detected (with a sudden increase in friction forces) was used as the measure of adhesion [9]. The scratch tests were realized on the smooth surface for the Vega implants in the inner surface. 

### 2.7. Micro-CT Evaluation

Similar to the SEM assessment, 16 samples were scanned by X-ray microtomography with a Bruker Skyscan 1272 microtomograph, at a resolution of 10 μm, with a 360° angular rotation, a spacing between images of 0.2°, and acquiring three images of each angle. After the complete scanning of the samples, the reconstruction of the images was carried out with the NRecon software version 2.0 (Bruker, Billerica, MA, USA), adjusting the alignment of the sample and the artifacts. 

The same software was used to perform measurements of the abutment–implant distances, measuring the distance 4 times in each reconstruction at 90° for each measurement.

The axial displacement was determined according to the formula:Axial Displacement = Final Distance − Initial Distance

### 2.8. Statistical Analysis: MICRO-CT

Statistical analysis was performed using Minitab 19 software, to determine statistically significant differences between the measurements of each study group sample, before and after each level of mechanical demand. The Kruskal–Wallis test was applied to determine if the median values of the 3 solicitations presented significant statistical differences, and if positive, the Mann–Whitney test was applied to evaluate differences between pairs of medians. In all cases, a confidence level of 95% was used, considering that the medians present statistically significant differences for a statistic value of *p* ≤ 0.05.

## 3. Results

### 3.1. SEM

An SEM is a type of electron microscope capable of producing high-resolution images of the surface of a sample. Because of the way images are created, SEM images have a characteristic three-dimensional appearance and are useful for evaluating surface structure samples. 

The study consisted of evaluating the presence of damage and deformations in the internal contact areas between abutment and implant, after subjecting the samples to different mechanical requests. 

For each of the 16 samples evaluated, a summary annex was prepared, which includes two representative SEM images (a, b) of the connection area of the dental implant, as well as another two representative SEM images (c, d) of the connection area of the dental abutment. The first two figures of each implant relate to the components in the manufacturing state. 

The main aspects observed by SEM microscopy for each of the 16 samples analyzed in this study are described below (Figure 2, Figure 3, Figure 4, Figure 5, Figure 6, Figure 7, Figure 8, Figure 9, Figure 10, Figure 11, Figure 12, Figure 13, Figure 14, Figure 15, Figure 16, Figure 17, Figure 18, Figure 19, Figure 20 and Figure 21).

#### 3.1.1. KL Implants

The main aspects observed by SEM microscopy for KL implants are described below:

**Figure 2 materials-17-00434-f002:**
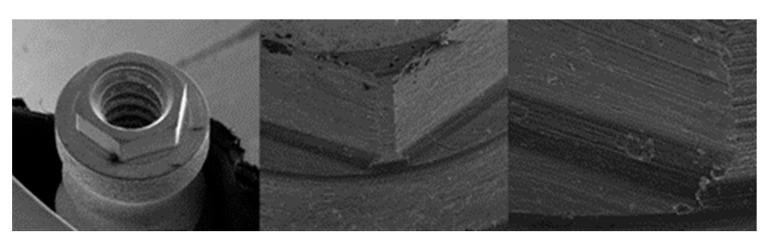
SEM micrographs of KL implants before the test with magnifications of 20×, 100×, and 300×, respectively, with the presence of multiple shot impacts of the hexagon produced during sandblasting.

**Figure 3 materials-17-00434-f003:**
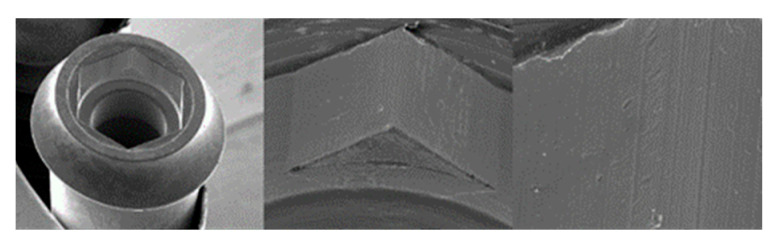
SEM micrographs of the KL abutments before the test with magnifications of 20×, 100×, and 300×, respectively, with details of machining marks and implant–abutment contact marks.

**Figure 4 materials-17-00434-f004:**
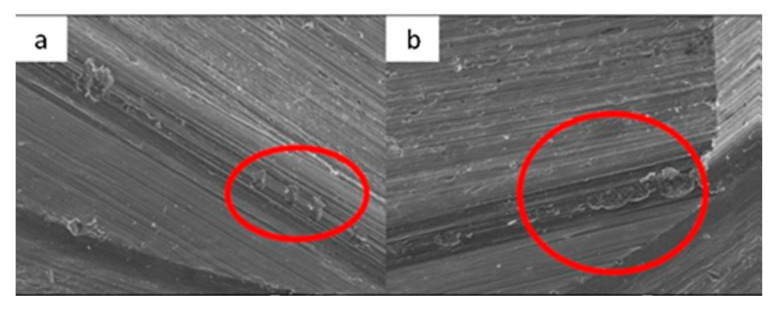
SEM micrographs of KL implants in phase I of the study with a 100× (**a**) and 300× (**b**) magnification with details of the presence of deformation marks (**a**) and abutment–implant contact marks on the faces of hexagons (**b**), signed by the red circles. SEM micrographs of KL abutments in phase I, with a 100× (**c**) and 300× (**d**) magnification, with details of abutment–implant contact marks on the faces of the hexagon, in the central area (**c**) and the upper area (**d**) corresponding to the implant–abutment contact area, signed by the red circles.

**Figure 5 materials-17-00434-f005:**
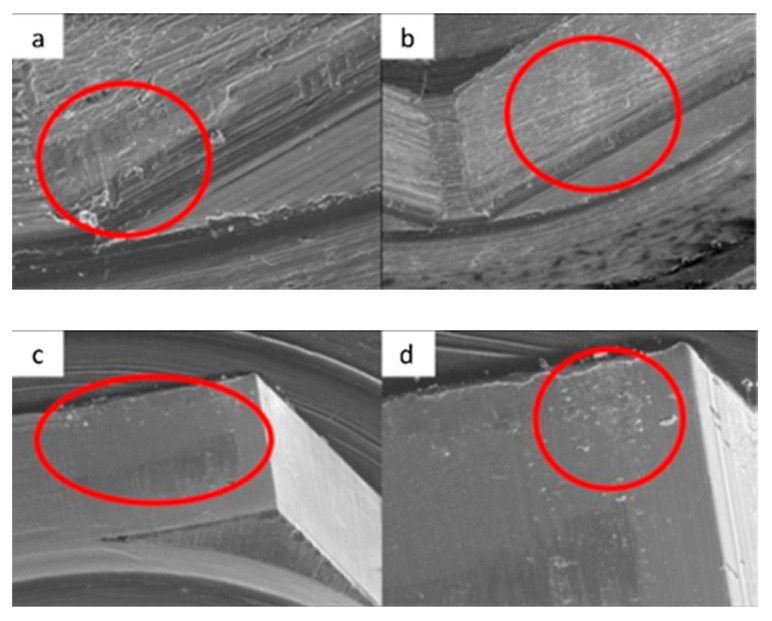
SEM micrographs of KL implants in phase II of the study with a 100× (**a**) and 300× (**b**) magnification, with the presence of deformations with signs of directionality (**a**) in the lower area of the faces of the base of the hexagon (**b**), signed by the red circles. SEM micrographs of the KL abutments in phase II of the study with a 100× (**c**) and 300× (**d**) magnification, with the presence of deformations with signs of directionality (**c**) in the lower area of the faces of the base of the hexagon (**d**), signed by the red circles.

**Figure 6 materials-17-00434-f006:**
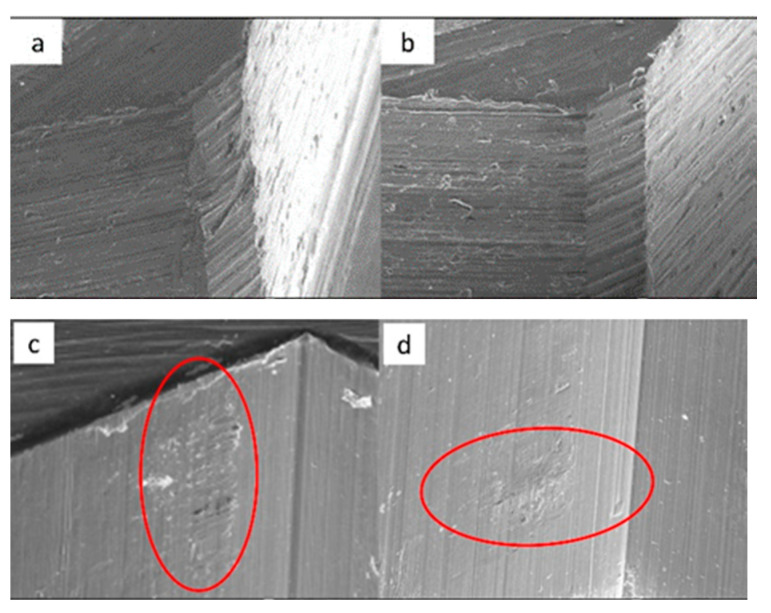
SEM micrographs of KL implants in phase III of the study with a 100× (**a**) and 300× (**b**) magnification, with the presence of deformation in the edge of the hexagon (**a**) and deformation of the upper plane (**b**). SEM micrographs of the KL abutments in phase III with a 100× (**c**) and 300× (**d**) magnification, with details of abutment–implant horizontal contact marks on the faces of the hexagon, signed by the red circles, with greater intensity on one of the faces (**c**).

#### 3.1.2. Essential Implants

The main aspects observed by SEM microscopy for Essential implants are described below:

**Figure 7 materials-17-00434-f007:**
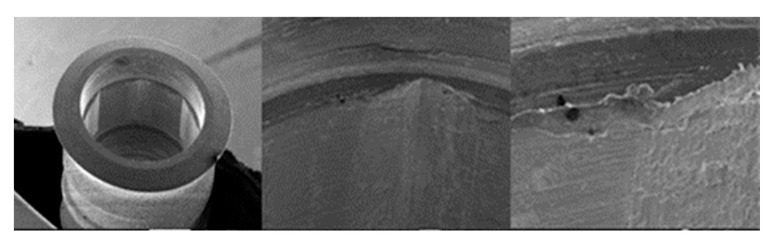
SEM images of ESSENTIAL Implants before the test at 20×, 100×, and 300× magnification, respectively.

**Figure 8 materials-17-00434-f008:**
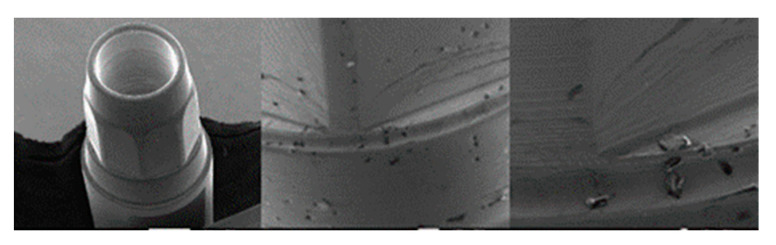
SEM images of ESSENTIAL abutments before the test at 20×, 100×, and 300× magnification, respectively.

**Figure 9 materials-17-00434-f009:**
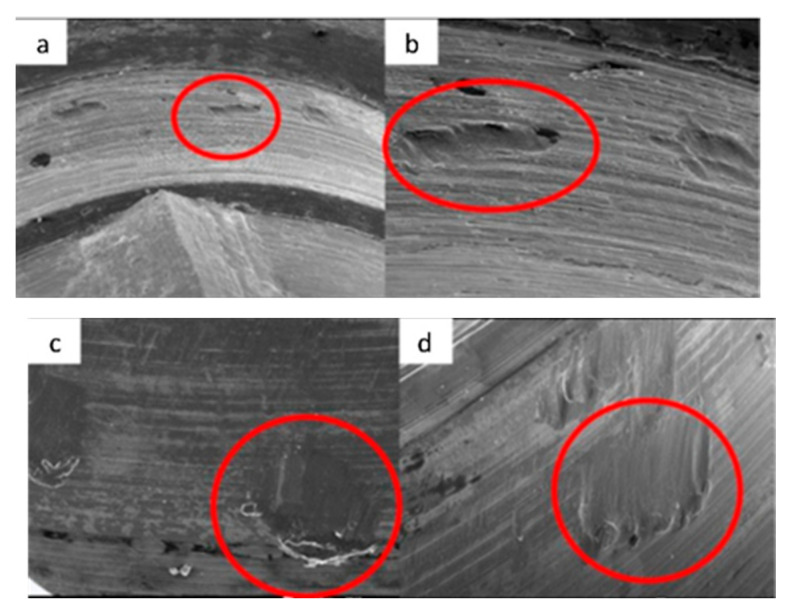
SEM micrographs of ESSENTIAL implants in phase I of the study with a 100× (**a**) and 300× (**b**) magnification, with details of deformation marks by contact in the upper conical plane, signed by the red circles. SEM micrographs of ESSENTIAL abutments in phase I of the study with a 100× (**c**) and 300× (**d**) magnification, with details of deformations caused by contact with vertical directionality, signed by the red circles.

**Figure 10 materials-17-00434-f010:**
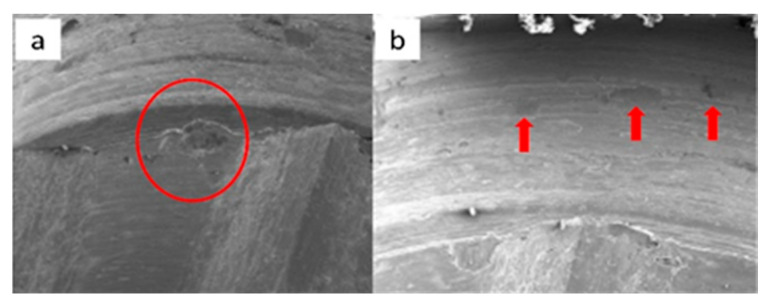
SEM micrographs of ESSENTIAL implants in phase II of the study with a 100× (**a**) and 300× (**b**) magnification, with details of deformations caused by contact in upper edge, signed by the red circle (**a**) and upper conical plane, signed by the red arrows, (**b**). SEM micrographs of ESSENTIAL abutments in phase II of the study with a 100× (**c**) and 300× (**d**) magnification, with details of deformations (contact marks) in the base of the hexagon (**c**) and in the lower conical plane (**d**), signed by the red circles.

**Figure 11 materials-17-00434-f011:**
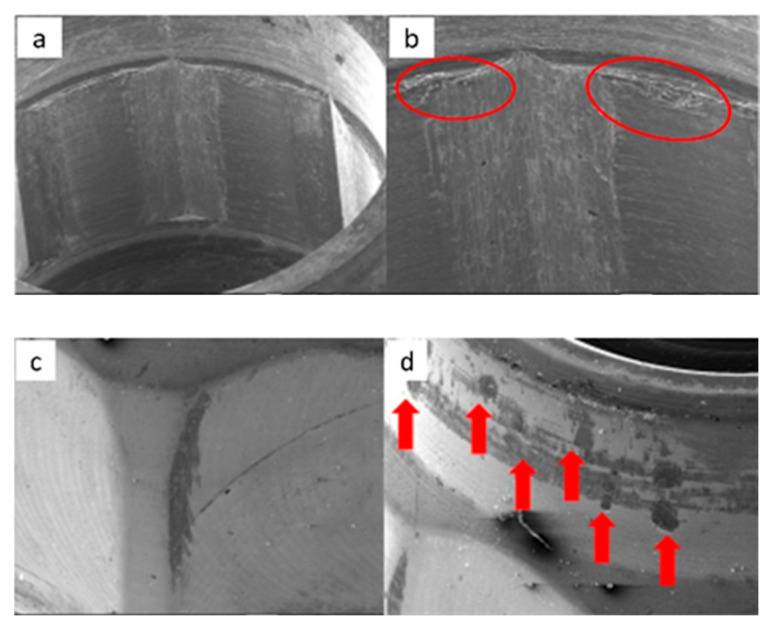
SEM micrographs of ESSENTIAL implants in phase III of the study with a 100× (**a**) and 300× (**b**) magnification, with details of deformations caused by contact on the top edges of the hexagon (**b**), signed by the red circles. SEM micrographs of ESSENTIAL abutments in phase III of the study with a 100× (**c**) and 300× (**d**) magnification, with contact marks on the hexagon face (**c**) and on the upper conical plane. Contact friction marks can also be seen on the edges of the polygon (**d**), signed by the red arrows.

#### 3.1.3. SK2 Implants

The main aspects observed by SEM microscopy for SK2 implants are described below:

**Figure 12 materials-17-00434-f012:**
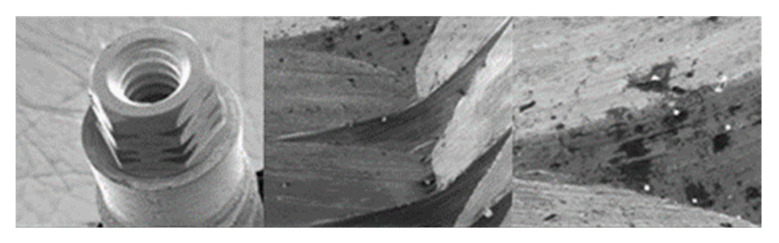
SEM images of SK2 implants before the test with magnifications of 20×, 100×, and 300×, respectively, and with details of the presence of burrs on the surface.

**Figure 13 materials-17-00434-f013:**
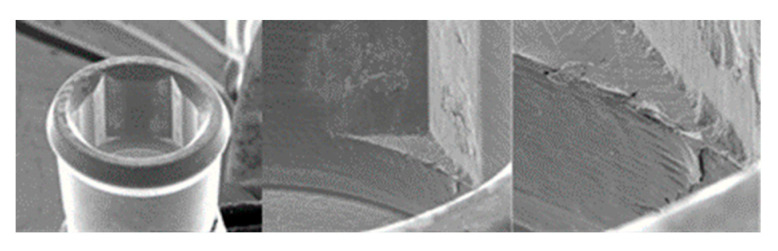
SEM images of SK2 abutments before the test with magnifications of 20×, 100×, and 300×, respectively, with details of machining marks.

**Figure 14 materials-17-00434-f014:**
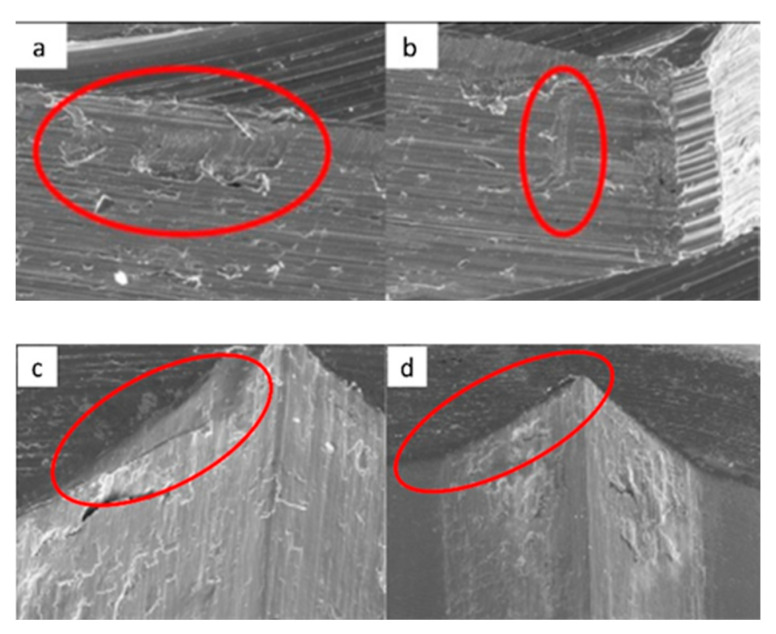
SEM micrographs of SK2 implants in phase I of the study with a 100× (**a**) and 300× (**b**) magnification, with details of deformation marks in the upper edge (**a**) and face of the hexagon (**b**), in addition to the presence of the previously verified manufacturing marks, signed by the red circles. SEM micrographs of SK2 abutments in phase I of the study with a 100× (**c**) and 300× (**d**) magnification, with details of deformations marks in the upper edges of the machining planes, signed by the red circles.

**Figure 15 materials-17-00434-f015:**
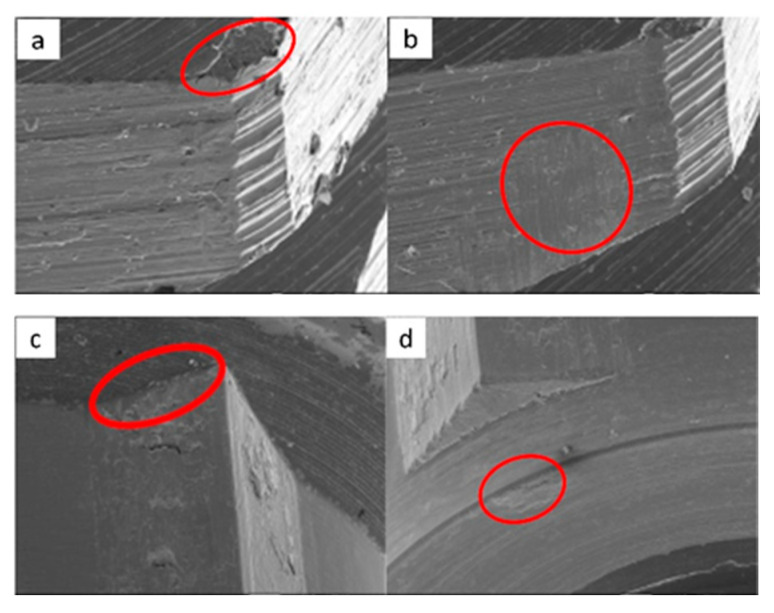
SEM micrographs of SK2 implants in phase II of the study with a 100× (**a**) and 300× (**b**) magnification, with details of deformations in the upper edge (**a**) and on the face of the hexagon (**b**), signed by the red circles. SEM micrographs of the SK2 abutments in phase II of the study with a 100× (**c**) and 300× (**d**) magnification, with details of deformation in the upper edge of the hexagon (**c**) and the lower conical plane (**d**), signed by the red circles.

**Figure 16 materials-17-00434-f016:**
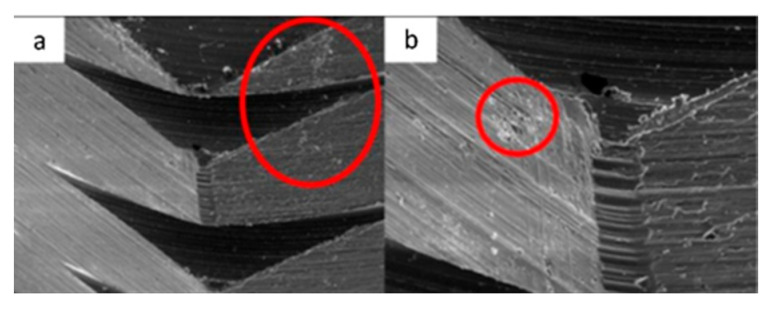
SEM microphotographs of SK2 implants in phase III of the study with a 100× (**a**) and 300× (**b**) magnification, with details of deformations on the edges (**a**) and faces of the hexagon (**b**), signed by the red circles. SEM microphotographs of SK2 pillars in phase III of the study with a 100× (**c**) and 300× (**d**) magnification, with details of deformations in the conical plane under the final plane of machining, signed by the red circle.

#### 3.1.4. Vega Implants

The main aspects observed by SEM microscopy for Vega implants are described below:

**Figure 17 materials-17-00434-f017:**
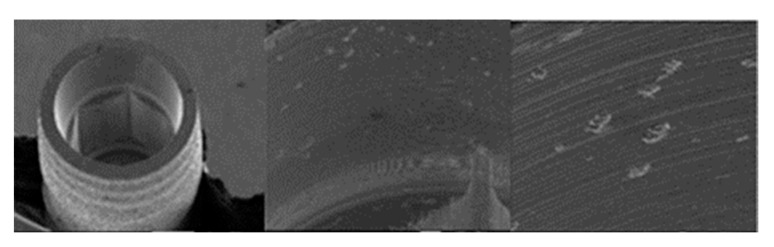
SEM images of VEGA Implants before the trial with magnifications of 20×, 100×, and 300×, respectively, and with details of the presence of burrs on the surface.

**Figure 18 materials-17-00434-f018:**
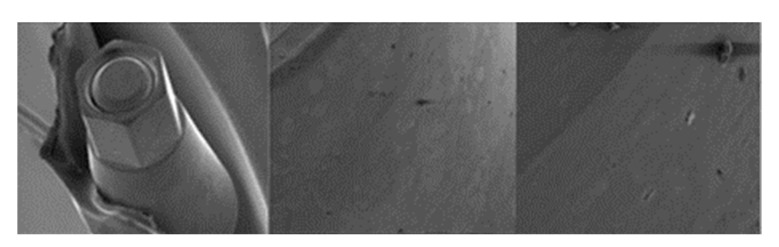
SEM images of the VEGA abutments before the trial with magnifications of 20×, 100×, and 300×, respectively, with details of machining marks.

**Figure 19 materials-17-00434-f019:**
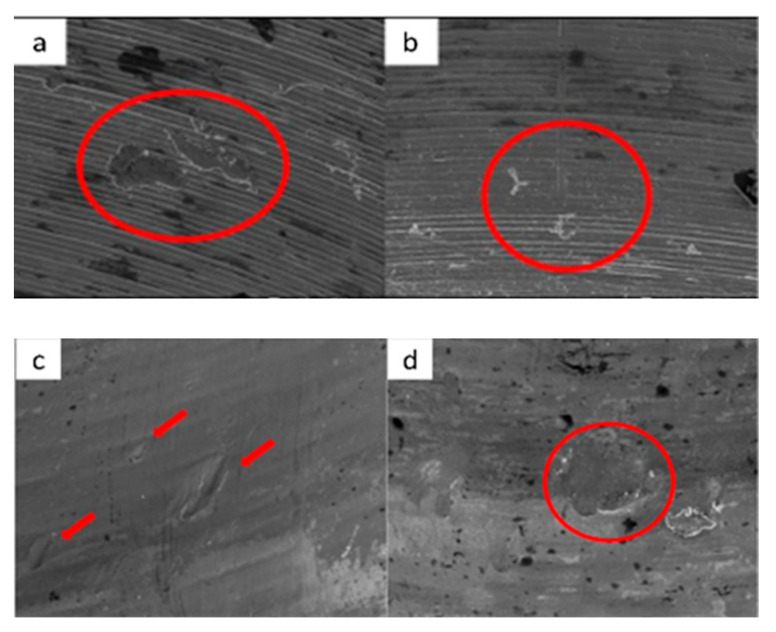
SEM micrographs of VEGA implants in phase I of the study with a 100× (**a**) and 300× (**b**) magnification, with details of deformed burrs (**a**) and contact lines in the conical plane (**b**), signed by the red arrows. SEM microphotographs with a 100× (**c**) and 300× (**d**) magnification, with details of multiple small deformations in the conical plane, signed by the red arrows (**c**) with a flattened appearance caused by contact (**d**), signed by the red circle.

**Figure 20 materials-17-00434-f020:**
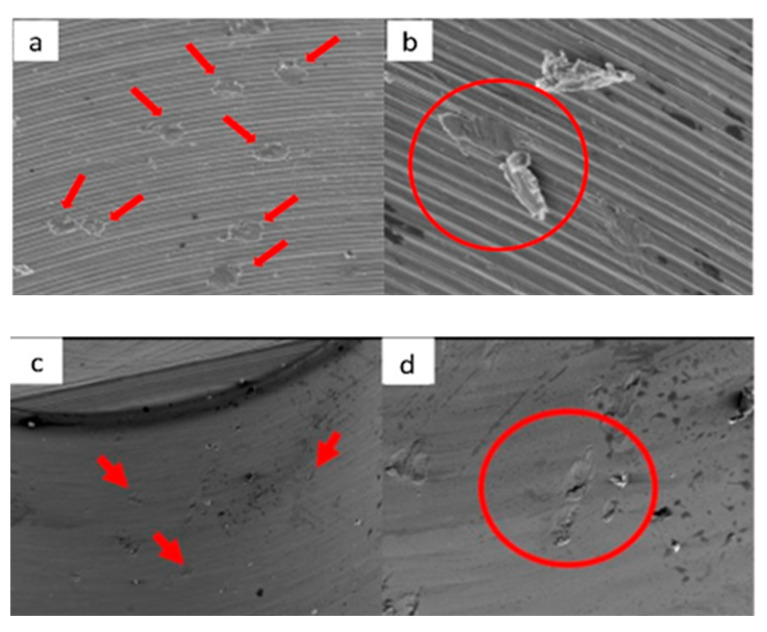
SEM microphotographs with a 100× (**a**) and 300× (**b**) magnification, with details of generalized deformations, signed by the red arrows (**a**) and the presence of burrs, signed by the red circle (**b**). SEM microphotographs with a 100× (**c**) and 300× (**d**) magnification, with details of multiple deformations in the conical plane, signed by the red arrows (**c**) and signs of vertical directionality caused by friction in the deformation marks, signed by the red circle (**d**).

**Figure 21 materials-17-00434-f021:**
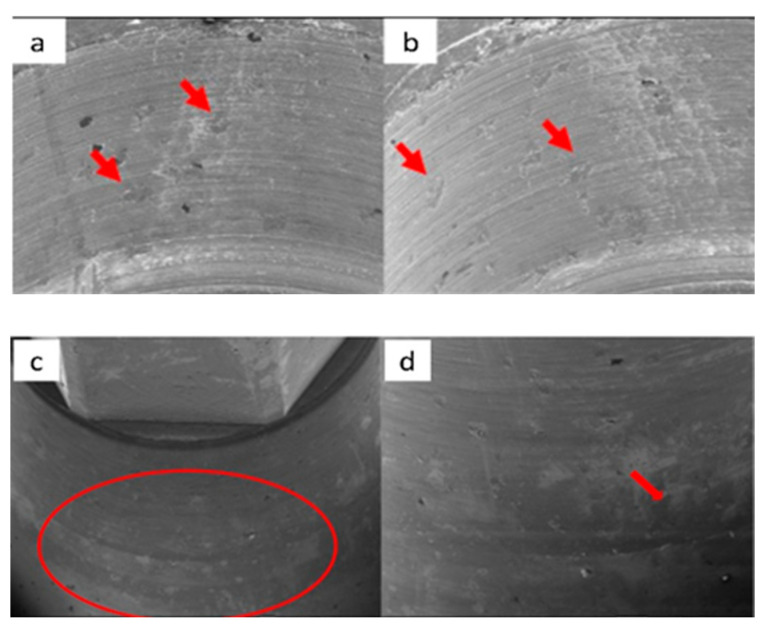
SEM microphotographs with a 100× (**a**) and 300× (**b**) magnification, detailing the presence of multiple deformations (**a**) with vertical directionality (**b**), signed by the red arrows. SEM microphotographs with a 100× (**c**) and 300× (**d**) magnification, with details of multiple deformations in the conical plane, signed by the red circle (**a**) and marked contact line, signed by the red arrow (**b**).

### 3.2. Residual Stress

Table 2 shows the residual stress values for each of the surfaces and for each of them in the different test conditions. It can be seen that the values are all negative, which means that they are residual compressive stresses of the surface. This is due to the fact that machining causes residual stress due to the removal of material from the titanium. It can be observed that the most negative value corresponds to Vega (V) since it is a dental implant which entire surface has been treated by shot blasting. These differences are statistically significant with *p* < 0.05 between all the surfaces and in all the test conditions. This process projects alumina particles at a pressure of 2.5 bar to cause roughening of the implant and higher residual stress. No significant differences are seen between K and SK. However, the E implants present intermediate values since they adsorb surface energy from the rough areas obtained by shot blasting, and in the areas close to the rough part there is a difference in residual stress. The residual stresses between E and the other implants also have significant differences. 

### 3.3. Scratch Test

The scratch test values can be seen in Table 2, and it can be observed that they are related to the compressive surface tension. The higher the residual surface tension, the harder the titanium becomes and therefore more force is needed to scratch the material. As in the case of the residual stress, it can be seen that the V implants have statistically significant differences in the scratching force with respect to E and E with respect to the two types of implants K and SK with the lowest values of residual stress. The scratch test values can be seen in Table 2, and it can be observed that they are related to the compressive surface tension. The higher the residual surface tension, the harder the titanium becomes and therefore more force is needed to scratch the material. As in the case of the residual stress, it can be seen that the V implants have statistically significant differences in the scratching force with respect to E and E with respect to the two types of implants K and SK with the lowest values of residual stress.

### 3.4. Micro-CT

Micro-CT is a 3D imaging technique that uses X-rays to see inside a cut object. Micro-CT, also called microtomography or microcomputed tomography, is similar to a CT scan, but on a small scale with much higher resolution. Sample images can be obtained with pixel sizes as small as 100 nanometers, and objects up to 200 mm in diameter can be scanned.

A total of 16 samples were evaluated by Micro-CT, one sample from each of the three study groups and a sample of each implant model in the manufacturing state. The selected sample was the one that obtained the highest tightening torque value, according to the previously obtained results. 

The reconstructions of the images resulting from Micro-CT were obtained, and two pillar–implant distances (a and b) were measured in the internal connection implants (Vega and Essential) and a distance (a) in the external connection implants (SK2 and KL) (Figure 22). Measurements of the abutment–implant distances were performed (Figure 23), measuring the distance four times in each reconstruction at 90° each measurement.

**Figure 22 materials-17-00434-f022:**
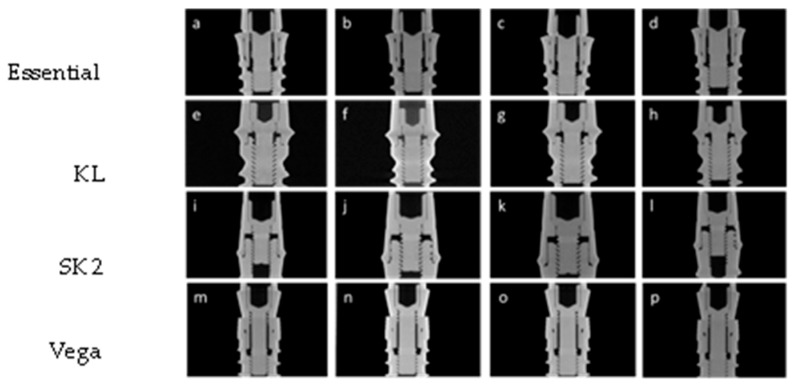
Plan of the reconstruction of the micro-CT analysis of the sample E0 (**a**), EU (**b**), EM (**c**), EMC (**d**); K0 (**e**), KU (**f**), KM (**g**), KMC (**h**); S0 (**i**), SU (**j**), SM (**k**), SMC (**l**); V0 (**m**), VU (**n**), VM (**o**), VMC (**p**).

**Figure 23 materials-17-00434-f023:**
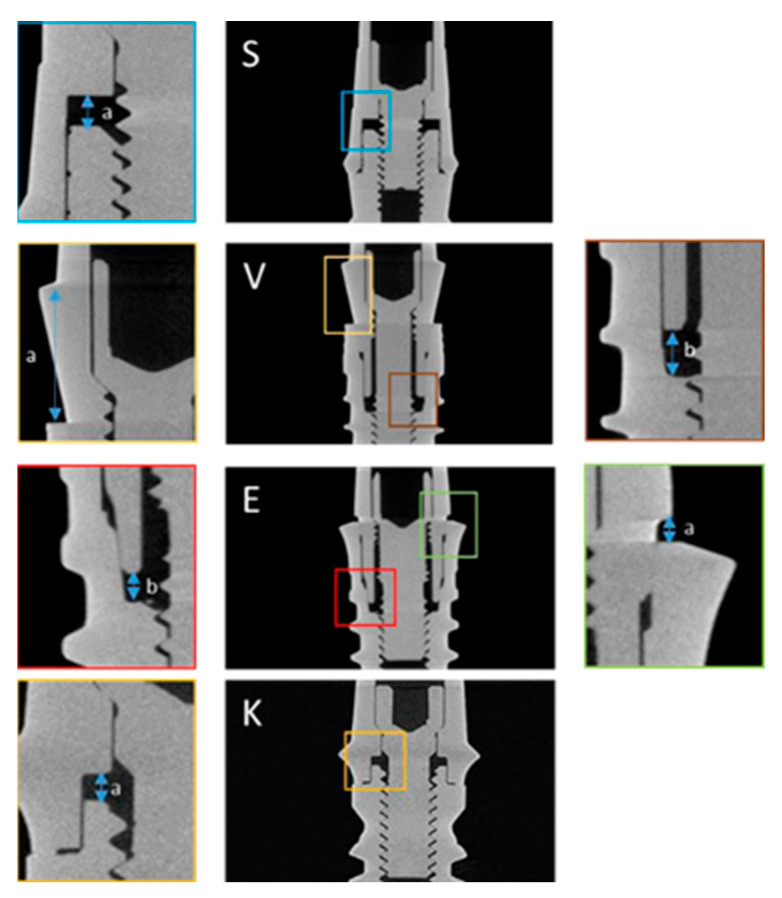
Distances measured for each of the implant models (S = SK2, V = Vega, E = Essential, K = KL).

Through statistical analysis, it was observed that implants SK2 and KL did not present statistically significant differences between the distances measured, before and after the application of the different mechanical requests. The Vega implant presented statistically significant differences in measure b, and the Essential implant presented statistically significant differences in both measures performed, a and b, respectively (Figure 24; Table 3).

**Table 3 materials-17-00434-t003:** Values of the p-statistic (probability) for each measure and implant model.

Model	Measure	*p*-Value
SK2	a	0.403
Vega	a	0.871
	b	0.020
KL	a	0.052
Essential	a	0.005
	b	0.008

**Figure 24 materials-17-00434-f024:**
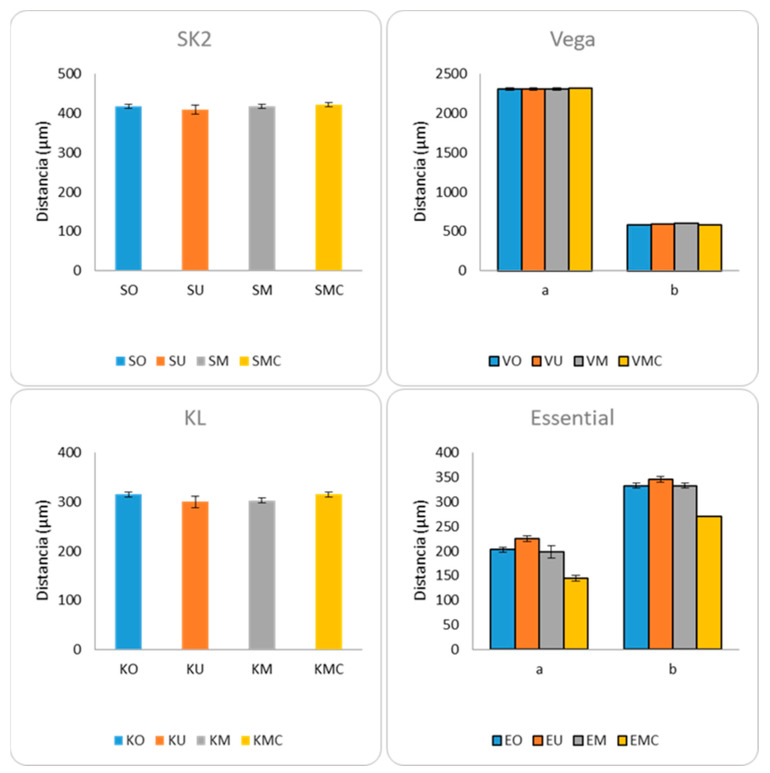
Measurements of abutment–implant distances for the four implant models tested.

Subsequently, a second statistical analysis of results between pairs of groups was carried out in the Vega and Essential groups, which had already presented statistically significant differences in distances measured. The differences between pairs of groups for the Vega implant, in distance b, were analyzed, and the results reflected the presence of statistically significant differences between VO and VM (*p* = 0.043) and between VM and VMC (*p* = 0.03). Regarding the Essential implant, statistically significant differences were observed between all pairs compared except EO and EM, for both measures a and b (Table 4).

## 4. Discussion

### 4.1. SEM Evaluation

The analysis of the 16 abutment–implant–screw assemblies by SEM aimed to determine the presence or absence of defects in the components, a product of the application of different mechanical requests.

SEM microphotographs, prior to our study, indicated that even a new precisely machined surface is not very smooth, an observation that agrees with the work of Tzenakis et al. [25] and Guzaitis et al. [26]. In the manufacturing state, all implants revealed minor structural alterations. Machining burrs (SK2, Vega), friction and contact marks (Essential), and strain marks (SK2 and KL) were observed. The respective abutments also showed minimal damage in the manufacturing state, probably produced by the machining process. All abutments, except SK2, showed a reduced number of slight contact deformation zones, the result of assembly between both components. These findings confirm that all machined surfaces are rough and explain the settling effect already described above. The settling effect is one of the factors contributing to the screw loss of preload, due to the distributed load on all metal interfaces [27]. Therefore, the initial tightening is used to smooth contact surfaces.

After single tightening, SK2 implants and abutments revealed the same aspects and marks observed in the manufacturing state, as well as areas of deformation not previously identified. By contrast, the remaining implants and abutments had the same deformation marks caused by contact, but with a slight increase in the number and size of the marks.

With multiple tightening, SEM analysis indicated an increase (number and dimension) in the damage in the form of marks or deformations. The Vega, SK2, and Essential implants/abutments also revealed friction marks with clear vertical directionality.

In the phase of multiple tightening followed by cyclic loading (500 N × 1000 cycles), SEM evaluation revealed the presence of the same deformation marks detected previously and an increase in the level of deformation, with a marked vertical direction. These marks, with the same morphology, detected in the Vega implant/abutment, are compatible with the relative displacement (intrusion) of the abutment towards the interior of the implant.

Several published studies corroborate our findings. SEM analysis was performed by Khraisat et al. [28] (EXT implants) and Tsuge et al. [29] (INT and EXT implants). These studies evaluated surface changes in the abutment screw thread and hexagonal corner of the implant, before and after loading, with 1 × 10^6^ cycles (Khraisat et al.) [25] and 2000 cycles (Tsuge et al.) [29]. Both studies showed minor damage to the surfaces of the pillar screw thread, after tightening, in control samples that were not loaded.

SEM was also conducted in the study by Cashman et al. [19] after 5 × 10^6^ loading cycles, to compare thread geometry and evaluate surface characteristics in INT connections. Differences in surface finish were visualized in the post-fatigue cycle, such as ductile delamination and rough surfaces.

Dental implants are made of pure titanium grade 3 or 4 because the mechanical properties are superior to those of titanium grade 1 and 2. The manufacturing companies made this change in titanium with higher grades because grades 1 and 2 had very little hardness and its plastic deformation was easy because the elastic limit was low [30,31]. There were many cases in which, in the placement of the implant in the mouth, the hexagonal connections were deformed by the tools and a good anchorage was not produced. In some cases, the so-called grade 5 is used, which is a Ti6Al4V alloy whose resistance values exceed 800 MPa, compared to the 450–600 MPa of pure titanium grades 3 and 4, and much higher than those of grades 1 and 2 with resistance values of 250–300 MPa [32,33]. It is for this reason that there are some companies that manufacture dental implants with the Ti6Al4V alloy in order to achieve higher mechanical strength, sacrificing some of the values of corrosion resistance, ion release to the medium, and osseointegration capacity. However, abutment–dental implant connection screws are generally made of the Ti6Al4V alloy because they are subjected to high mechanical stress and the properties of pure titanium would be small for the values that the screws undergo in service [33].

Titanium surface defects should be reduced as they are points at which a fatigue crack can initiate. These defects are points of concentration of mechanical stresses that can initiate a crack. The mechanical cycles of chewing lead to the propagation of this crack and can lead to the breakage of the dental implant. These defects are more dangerous when they are located at the connection points of the abutment implant, where the resistant section is smaller [34].

One of the advantages of VEGA implants is that they present a rough surface over the entire surface generated by shot blasting. This procedure of producing roughness is achieved by spraying abrasive particles onto the surface at a pressure between 2 and 4 MPa. The shot blasting process causes the particles to remove defects and seal possible cracks in the surface. In addition, shot blasting generates compressive stress that inhibits or at least delays the formation of a fatigue crack and ensures good long-term mechanical behavior. It is well known that cracks require tensile stresses for crack propagation [35,36].

It can be seen that the mechanical cycles for SMC, VMC, KMC, and EMC implants have more residual stress and scratching force values than the S, V, K, and E control implants because the mechanical cycles to which the dental implant is subjected increase the accumulated damage to the dental implant. In other words, fatigue causes an increase in deformation, which leads to an increase in the internal stress values of the implant and an increase in the hardness, and therefore in the scratch resistance. It is for this reason that these differences due to mechanical cycling (fatigue) are also seen in the samples with multiple tightening but to a lesser extent.

The implants with internal connection also have stress produced by the abutment and therefore have lower levels of deformation on their surface compared to those with external connection.

### 4.2. Micro-CT Evaluation

This phase focused on the analysis of 16 implant–abutment assemblies by X-ray microtomography, to determine the possible changes in the distances between the abutment and the implant, after different mechanical demands.

In the literature, we can find that cyclic loading causes displacement of the abutment in relation to the implant—defined as implant–abutment micro-movement—and can alter the size of the microgap [37,38], cause wear [39,40], screw loosening [41], and peri-implant bone loss compromising the clinical longevity of the implant [4,42].

Our study has shown that for EXT connection implants (SK2 and KL), the distance measured between abutment and implant in the different mechanical demands did not present significant differences. In contrast, after multiple tightening and cyclic loading, INT connection Vega implants presented significant differences in one (measure b) and Essential implants, in the two distances evaluated.

At a general level, a clear intrusion of the abutment into the implant could only be confirmed in the Essential implant model, which had obtained its maximum value under Phase III conditions (multiple pre-load + cyclic loading). These findings from the Mico-CT analysis corroborate the results of our SEM assessment, which demonstrated levels of abutment intrusion within the implant in Vega and Essential sample groups. This vertical displacement in the axial axis occurs in all clinical and laboratory stages of implant-supported prostheses. The application of different torque during the work process and the appearance of discrepancies in the position of the abutment result in the loss of passive adjustment in the suprastructure, which ultimately leads to a mismatch of the implanted prosthesis [36].

Compared with the EXT connection, the INT connection has a wedge effect. In this effect, the abutment acts as a wedge, concentrating the axial compressive force in the direction of abutment insertion, which increases contact pressure and frictional resistance [6,11]. In addition, as the tightening torque increases, the wedge effect of the abutment increases.

It is considered that the axial displacement of the abutment, under functional load, can have a great effect on the stability of the screw joint of the INT connection. When using the abutment, which lacks a vertical stop, axial displacement of the abutment can occur under functional load in the oral cavity. This implies that the tensional force inside the screw decreases, indicating a loss of preload. When preload decreases, it reaches a critical point that under a load causes screw loosening [15].

In the Lee et al. study, after the cyclic loading of 250 N × 10^5^ of the final prosthesis, some amount of axial displacement was found in the INT and EXT groups. Similar to our study, the INT connection showed a more severe axial displacement. Compared with the EXT connection, the INT group has a conical shape and a wider coupling surface area, which could lead to a greater sedimentation effect [9]. The importance of coupling differences between different types of implant connections is due to the passive fitting of prostheses and the avoidance of screw loosening [43]. The coupling effect of the abutment can occur and be attributed to axial displacement, at the same time as tightening procedures are repeated. As the tightening torque is applied, the abutments of the INT connection could fit into the implant. Without the defined vertical stop, the conical portion of the abutment suffers intrusion into the hollow area, which is inside the implant, resulting in axial displacement [6]. This effect may occur in two clinical stages: first during screw tightening and second during occlusal loading.

Our in vitro study suggested a mechanical advantage for the INT connection, because the fixation and stability of this connection are not the sole responsibility of the screw, but rather conferred by the frictional resistance, resulting from contact between the conical mating part of the abutment and the implant. The application of axial compressive forces seems to enhance this frictional resistance. However, this long-term axial displacement, in INT conical connection implants, presents clinical implications and should be managed clinically, with provisional restorations, to avoid future negative occlusion. Some authors advise using a provisional prosthesis for a sufficient period of time and performing the definitive restoration after axial displacement has become self-limited [44].

The manufacturing tolerance between the different samples evaluated of the same implant model could be counteracting the axial displacement values obtained in our study. In order to eliminate this uncontrolled variable in future studies, we believe it is advisable to carry out the studies sequentially, with the same set of samples for each implant model to be evaluated. In this way, by scanning implant–abutment, both in the manufacturing state and then in each different mechanical demand, we will have an accurate determination of the level of impact caused by each mechanical request, without the interference of possible manufacturing tolerances.

There were also some limitations to this study related to the in vitro condition. Similar to other studies biological confounding factors related to the bone–implant interface (bone deformation or fracture, and loss of mechanical stability) tend to dwarf the true effect of static and cyclic loading [45]; therefore, there is no generalization from the in vitro to the clinical phase in the present experimental conditions used [46].

For future studies, a bone model support, for the samples, with similar characteristics to human alveolar bone should be considered. The oral environment and conditions could not be exactly simulated (saliva, Ph, biofilm, temperature), and thus the results should be interpreted with caution and validated in a clinical condition. There are variables that can be explored and others that can be introduced in future studies following this line of research, such as the application of eccentric forces, angulated abutments, larger compressive forces during more cycles, different abutments and screws materials, testing lubricants, etc.

## 5. Conclusions

Within the limitations of this in vitro study, we can draw the following conclusions:

Implant–abutment samples, in the manufacturing state, revealed minor structural changes and minimal damage from the machining process. 

SEM analysis of external connection implants did not show clear signs of axial displacement. This type of connection, characterized by an external hexagon and a horizontal plane of abutment/implant support, would greatly avoid the intrusion of the abutment into the implant. The higher the residual stress, the higher the scratching force and the lower the plastic deformation in the tightening process. SEM analysis of internal connection implants revealed small levels of axial displacement of the abutment inside the implant. The internal conical type of connection would allow and facilitate the intrusion of the abutment into the implant. The frictional deformation marks, with vertical direction, reflected the intrusion of the abutment into the implant.

The implant–abutment distances evaluated by micro-CT did not show statistically significant differences in external connection samples, for any of the mechanical requests. A clear axial displacement of the abutment into the implant was only confirmed in the internal connection Essential model, which obtained its maximum value under multiple tightening and cyclic loading. Clinically, this situation should be taken into account to avoid future negative occlusion in final restoration.

## Figures and Tables

**Table 2 materials-17-00434-t002:** Residual stresses of the different surfaces and mean scratch values of the different titanium surfaces. Values are mean ± standard deviation. Statistical differences vs. smooth surfaces for each column are indicated by single and double asterisk-symbols (*p* < 0.05). Values without asterisk show statistically significant differences with those with a single or double asterisks. There are also significant differences between single and double asterisks.

Samples	Residual Stress (MPa)	Scratch Force (mN)
S	−6.2 ± 1.3	159 ± 23
V	−201.4 ± 14.3 *	299 ± 18 *
K	−5.8 ± 0.9	150 ± 20
E	−25.2 ± 6.2 **	198 ± 29 **
SU	−6.0 ± 1.2	157 ± 20
VU	−200.3 ± 10.0 *	307 ± 19 *
KU	−5.5 ± 1.9	151 ± 32
EU	−33.1 ± 7.2 **	190 ± 25 **
SM	−5.9 ± 1.0	179 ± 24
VM	−221.2 ± 32.0 *	312 ± 28 *
KM	−6.3 ± 0.8	170 ± 20
EM	−37.2 ± 6.2 **	201 ± 11 **
SMC	−8.2 ± 1.9	189 ± 30
VMC	−241.1 ± 14.3 *	350 ± 38 *
KMC	−10.8 ± 2.1	185 ± 32
EMC	−45.2 ± 7.8 **	218 ± 19 **

**Table 4 materials-17-00434-t004:** Comparison between pairs of values for the b measure of the Vega implant and the two measures of the Essential implant.

Distance b	*p*-Value	Distance a	*p*-Value	Distance b	*p*-Value
VO vs. VU	0.312	EO vs. EU	0.025	EO vs. EU	0.047
VO vs. VM	0.043	EO vs. EM	0.739	EO vs. EM	1
VO vs. VMC	0.386	EO vs. EMC	0.025	EO vs. EMC	0.023
VU vs. VM	0.058	EU vs. EM	0.027	EU vs. EM	0.047
VU vs. VMC	0.083	EU vs. EMC	0.027	EU vs. EMC	0.025
VM vs. VMC	0.03	EM vs. EMC	0.027	EM vs. EMC	0.023

## Data Availability

The data presented in this study are available on request from the corresponding author.

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
