# Peer review of "SEM Analysis and Micro-CT Evaluation of Four Dental Implants after Three Different Mechanical Requests—In Vitro Study"

_materials, 2024, doi:10.3390/ma17020434_

Round 1
Reviewer 1 Report
Comments and Suggestions for Authors
The authors used scanning electron microscopy and micro-CT to study the implant-abutment distances for several commercial models to determine any manufacturing defects. However, the manuscript does not discuss any material aspects, hence, it is not suitable for this journal.
Author Response
The article has been adapted, also characteristics of materials employee has been added in the design of the study to make it suitable for publication in this journal.
Reviewer 2 Report
Comments and Suggestions for Authors
It is an honor to review this article and I would like to congratulate with authors for all the effort that they did to conduct this in vitro study. The topic is interesting and appreciable from a scientific point of view, and it could be a valid point to start new clinical studies. The present in-vitro study is well designed and conducted, and the manuscript is quite clear.
There are some comments below.
The English is clear and spelled correctly.
Abstract: The abstract correctly summarizes the study design and purpose as the title as well.
Keywords: The keywords are correct and perfectly fitting the study design, I would add “Fatigue test” and "mechanical stresses".
Introduction: The introduction is well organized and clear. It has several references, but I suggest to add other appropriate ones. LINE 40-43: When you talk about micro-movements and micro-gap of the implant-abutment interface and also in further sentences you should consider that the neck of the implant and the implant-abutment interface may have a role in mechanical stress, please cite the two following recent articles that are very similar in concepts with your view:
Cosola, S.; Toti, P.; Babetto, E.; Covani, U.; Peñarrocha-Diago, M.; Peñarrocha-Oltra, D. In-vitro fatigue and fracture performance of three different ferrulized implant connections used in fixed prosthesis. J. Dent. Sci. 2021, 16, 397–403.
Cosola, S.; Toti, P.; Babetto, E.; Covani, U.; Peñarrocha-Diago, M.; Peñarrocha-Oltra, D. In-Vitro Investigation of Fatigue and Fracture Behavior of Transmucosal versus Submerged Bone Level Implants Used in Fixed Prosthesis. Appl. Sci. 2021, 11, 6186.
Materials and Methods: The methodology is well described and complete. The sample size and statistical procedures are correct. The study has the aim to investigate the integrity and wear of implant components and evaluating the axial displacement of the implant-abutment assembly by Micro-Ct and using SEM 20, in different type of implant connections after 3 distinct mechanical requests. I have some doubts regarding bias. For example, where the implants were placed, a cancellous bone with a surrounding 2 mm thickness cortical bone is quite hard compared to the normal bone or a bone substitute, the mechanical test may depend on the density and the quality of the bone model used?
Results: The results are clear and supported by an adequate number of figures and tables (suggested to be more professional, ok with colors). The statistical analysis is correct.
Discussion and Conclusions
In the limitation of the study you should discuss about my concerns reported before in the materials and methods and also about other limitations, for example the short sample of implants, the brand or the fact that it is an in-vitro study and in the oral cavity it could be more complex.
I agree with the discussion and conclusion but please, when you talk about implant connections add references between LINE 431 and 436 talking about also what can happen in clinical situations and real dentistry:
D'Orto B, Chiavenna C, Leone R, Longoni M, Nagni M, Capparè P. Marginal Bone Loss Compared in Internal and External Implant Connections: Retrospective Clinical Study at 6-Years Follow-Up. Biomedicines. 2023 Apr 8;11(4):1128. doi: 10.3390/biomedicines11041128.
Please make the conclusion a bit softer.
Author Response
Thank you very much for your time and dedication. Bibliographic references have been added and bibliography has been improved. The discussion and conclusions have been modified thanks to the valuable feedback of the reviewer.

Reviewer 3 Report
Comments and Suggestions for Authors
Abstract:
“Abstract: Statement of problem: Implant supported rehabilitations are an increasingly frequent 16 practice to replace lost teeth. It would be desirable that prior to their clinical application, all implant 17 components had to demonstrate suitable durability in laboratory studies. For that purpose, fatigue 18 tests, utilizing cyclic loading, typically simulate masticatory function in vitro.” This paragraph should be removed: usually the abstract is concise.
“and to access the axial displacement” maybe it should have been “and to assess the axial displacement”
Introduction:
“In the INT connection”, “Compared with the EXT connection”: the first time an abbreviation is used, its meaning should be declared.
Materials & Methods:
Table 1 can be removed.
Typo error: “2.5. Mico-CT Evaluation” should be “2.5. Micro-CT Evaluation”
Results:
Please consider to submit as supplemental materials most of the pictures from Figure 2 to figure 33.
Figure 36. Typo error: “Distancia” should be “Distance”
Comments on the Quality of English LanguageThe manuscript would benefit from a native English speaker language revision: few sentences meaning is hard to understand
Author Response
In the abstract, the suggested modification has been made.
The meanings of the terms INT and EXT have been added in the introduction the first time that have been mentioned.
We consider that Table 1 is necessary for the context of the study. Even so, if the reviewer considers it is unnecessary, we can remove it.
Furthermore, other grammatical mistakes have been solved.

Round 2
Reviewer 1 Report
Comments and Suggestions for Authors
There is still no results and discussion related material science. For example, no processing of materials is discussed that can lead to deviation or defects in the implants.
The content of the manuscript is more suitable for a manufacturing or measurement science journal.
Author Response
Dear Editor and Reviewer, we apologize for not having been clear in the changes made to the manuscript regarding the materials characteristics. Some paragraphs were added referring to the material used in the material and method and discussion.
Also, 7 new bibliographical references (26-32) were added to enrich the work and all the text added to the manuscript is now underlined. We hope it is now suitable for your journal.